# Source Identification and Genome-Wide Association Analysis of Crown Rot Resistance in Wheat

**DOI:** 10.3390/plants11151912

**Published:** 2022-07-24

**Authors:** Lefan Pu, Farhan Goher, Mengke Zeng, Dongsheng Wu, Qingdong Zeng, Dejun Han, Chunlian Li

**Affiliations:** 1State Key Laboratory of Crop Stress Biology in Arid Areas, College of Agronomy, Northwest A&F University, Xianyang 712100, China; lefanpu@163.com (L.P.); ZengMengKe0370@163.com (M.Z.); 17725124748@163.com (D.W.); 2State Key Laboratory of Crop Stress Biology for Arid Areas, College of Plant Protection, Northwest A&F University, Xianyang 712100, China; goherfarhan@nwafu.edu.cn (F.G.); zengqd@nwafu.edu.cn (Q.Z.)

**Keywords:** wheat, crown rot, resistance source identification, genome-wide association analysis

## Abstract

Crown rot (CR) is a soil-borne disease of wheat in arid and semiarid areas of the world. The incidence rate and severity of CR are increasing with each passing year, which seriously threatens the safety of world wheat production. Here, 522 wheat varieties/lines representing genetic diversity were used to identify and evaluate the resistance source to CR disease. Six varieties, including Zimai 12, Xinong 509, Mazhamai, Sifangmai, and Dawson, were classified as resistant ® to CR. Seventy-nine varieties were classified as moderately resistant (MR) to CR, accounting for 15.13% of the tested varieties. The wheat 660 K SNP array was used to identify resistance loci by genome-wide association analysis (GWAS). A total of 33 SNPs, located on chromosomes 1A, 1B, 1D, 4A, and 4D, were significantly correlated with seedling resistance to CR in two years. Among them, one SNP on chromosome 1A and nine SNPs on chromosome 1B showed most significant resistance to disease, phenotypic variance explained (PVE) by these SNPs were more than 8.45%. Except that significant locus AX-110436287 and AX109621209 on chromosome 1B and AX-94692276 on 1D are close to the already reported QTL, other SNPs are newly discovered resistance loci. These results could lay a strong theoretical foundation for the genetic improvement and breeding for CR resistance in wheat.

## 1. Introduction

In recent years, crown rot (CR) caused by a variety of Fusarium species has become a serious disease of wheat, with *Fusarium pseudograminearum* being the most prevalent and destructive species [1,2,3,4]. CR is common in arid and semiarid regions across the world [5], posing a severe threat to global wheat yields and food security. This disease has rarely documented in Chinese history, but its incidence and prevalence have been rising year after year since 2012, clearly posturing a new challenge to China’s wheat production [6,7,8,9]. In addition to reducing yield, CR also leads to the potential threat of mycotoxin contamination, such as deoxynivalenol (DON) accumulated in stems. *F. pseudograminearum* can survive in wheat stubble for up to three years [10]. Therefore, the increasing use of no-tillage and stubble retention to maintain soil moisture leads to an increase in the incidence of CR in many regions of the world [11]. Generally, the wheat plants infected by *F. pseudograminearum* appear as dark brown stripes or spots at the stem base and spread upward along the leaf sheath and stem [12,13]. If the temperature and humidity are appropriate, internodes from infected wheat plants will show pinkish mycelial growth of *F. pseudograminearum* [12,13].

Usually, techniques such as appropriate rotation, stubble management, limiting straw returning to the field, and regulating the quantity of pathogen are used to prevent and control CR, but these measures do not eliminate the problem. Breeding and utilizing resistant germplasms are the most effective and environmentally friendly ways to prevent and control diseases. However, no wheat germplasms that are immune or highly resistant to CR have been found, and only a few varieties are moderately resistant to CR. Most wheat germplasms with moderate resistance to CR come from Australia and the United States, such as Janz, Gluyas early, Sunco, 2-49, Baxter, Lang, Kukri, Ernie, CSCR6, Kennedy, and Wollaroi [14,15,16,17,18], but these germplasms are old varieties with relatively low yield. Liu et al. identified 2514 different types of wheat germplasms from all over the world and found that late-maturing or winter wheat is more resistant to CR [16]. Yang et al. conducted greenhouse seedling identification of 234 wheat varieties in the Huang-Huai wheat-growing region in China, and found no immune and highly resistant varieties; only seven varieties were shown to have resistance to CR [7]. Shi et al. identified 205 varieties/lines, and found that only three varieties, LS4607, Cunmai 633, and Pubing 01, have good resistance to CR [19]. Jin et al. identified 670 wheat germplasms resistant to CR at seedling stage and found that only a few varieties/lines have the same resistance as the resistant control Sunco, while most varieties/lines showed high sensitivity [20]. Pariyar et al. identified 161 wheat germplasms representing genetic diversity and different geographical regions for CR resistance under different environments, and found that only a few varieties have moderate resistance to CR, and there are significant differences in variety resistance and resistance loci under different environments, indicating that genotype and environment interaction had a great impact on resistance to CR [21].

QTL mapping of resistance is the basis of exploration and utilization of resistance genes. At present, more than 90 QTLs related to CR resistance have been found on 19 chromosomes of wheat, and the QTLs on chromosome 1AS (*Xbarc148-Xgwm164*), 2BS (*Xcfa2278-Xgdm86*), 3BL (*Xgwm299-Xwmc274*), 4BS (*Xgwm165-Xwmc467*), 1DL (*Xwmc216-Xcfd19*), 2DL (*Xcfd73*), 5DS (*Xcfd78*), and 6DL (*Xbarc273-Xbarc196*) were detected in multiple environments and different populations [22,23,24,25]. Resistance source 2-49 carries the most resistance genes, such as QTL on chromosomes 1AL, 1AS, 1BS, 1DL, 3BL, and 4BS [22,23,24,25]. Resistance QTLs from Sunco are mainly located on chromosomes 2B, 2DS, 3BS, 4BS, 4D, and 7A [23,24]. Among them, the major QTL located on 3BL between *Xgwm299* and *Xwmc274* was found in at least three populations. 

The genome-wide association study (GWAS) is a bioinformatics approach based on linkage disequilibrium (LD), and the improvement of association analysis techniques for whole genome resequencing (WGR) has improved GWAS in recent years by combination of the complex traits of populations with millions of high-density single nucleotide polymorphisms (SNPS) in large populations, to screen for molecular markers significantly linked with phenotypic variation and investigate their genetic effects [26]. GWAS is less time-, manpower-, and resource-intensive than conventional chain analysis. In the early 2000s, GWAS was gradually applied to plant diseases, as initially functional to the study of human diseases. Nowadays, association analysis has been applied in the localization of many wheat resistance genes, such as wheat *Stagonospora nodorum* blotch [27], Russian wheat aphid [28], stripe rust [29], etc. With the development of whole genome resequencing (WGR) technology, GWAS has been more and more applied to the discovery of CR resistance gene loci. Yang et al. used wheat 660 K chip and GWAS technology, and found 266 SNPs related to CR resistance in the 7.0 MB region of 6A chromosome; the region contains 51 annotation genes [7]. Pariyar et al. identified 15 QTLs related to wheat CR resistance differentially expressed in different growth environments on chromosomes such as 2AL, 3AS, 3BS, 3DL, 4BS, 5BS, 5DS, 5DL, 6BS, 6BL, and 6DS by GWAS [21]. Jin et al. analyzed the CR resistance loci of 358 Chinese wheat germplasms by GWAS, and found 104 SNPs on chromosomes such as 1BS, 1DS, 2AL, 5AL, 5DS, 5DL, 6BS, and 7BL [20]. GWAS analysis of 311 wheat varieties/lines showed that 6 SNPs were associated with seedling resistance on chromosomes 1A and 6B, and 15 SNPs were associated with adult plant resistance on chromosomes 1A, 1B, 3A, 3B, 6A, and 6D [30]. Yang et al. mapped the QTL to chromosome 4B by GWAS of introgression lines population of Yanzhan 1, and then the QTL was located in a 0.53 Mb region by haplotype analysis and QTL mapping of a double haploid (DH) population [31]. Through virus-induced gene silencing (VIGS), tetraploid mutant, and AK58 mutant library, they verified that the candidate gene was the negative regulator to CR, which provide an effective theoretical basis for CR resistance breeding [31]. GWAS is widely utilized to explore plant disease resistance loci of different germplasms. The combination of GWAS, QTL, and meta QTL will be the strong basis to further improve the fine mapping, gene cloning, and genetic linkage marker development of disease resistance genes in the future.

In this paper, 522 wheat varieties/lines were evaluated for CR resistance, and genome-wide association studies (GWAS) were performed using the wheat 660 K genotyping assay. This study provides information for utilization of CR-resistance wheat germplasm and further exploration of disease resistance related genes, and lays a foundation for genetic improvement and variety breeding for wheat resistance to CR.

## 2. Results

### 2.1. Phenotypic Analyses

The ANOVA (analysis of variance) showed that there were significant differences in Disease index (DI) among the tested varieties/lines under different sowing-time conditions (*p* < 0.001), but the difference between replications was not significant (Table 1). In addition, the distribution of DI in the tested varieties/lines showed that the DI was concentrated between 20 and 60 (Figure 1), and the resistance of the tested varieties/lines to CR showed a continuous distribution, indicating that the resistance of wheat to CR was controlled by polygenes. 

To exclude the influence of the environment on the phenotypic results of the seedling stage, we recorded the weather conditions in the test month and found that the climate conditions in March 2019 and March 2020 were the same. The statistics are as follows: Synopsis of weather statistics of Yangling in March 2019: monthly maximum temperature: 26 °C (appearing on 20 March 2019), monthly minimum temperature: 0 °C (appearing on 4 March 2019) (daytime), sunny days: 6 days; rain: 5 days; cloudy and overcast days: 24 days; snow: 0 days. (Night) sunny days: 3 days; rain: 2 days; cloudy and overcast days: 24 days; snow: 0 days. Synopsis of weather statistics of Yangling in March 2020: Monthly maximum temperature: 25 °C (appearing on 25 March 2020), monthly minimum temperature: 0 °C (appearing on 2 March 2020) (daytime) sunny days: 5 days; rain: 4 days; cloudy and overcast days: 22 days; snow: 0 days. (Night) sunny days: 6 days; rain: 3 days; cloudy and overcast days: 22 days; snow: 0 days.

### 2.2. Evaluation of Resistance to CR in Wheat Cultivars

Based on the screening results of two years, the tested varieties/lines can be divided into four categories of disease rating scale: resistant (R, 0 < DI ≤ 15), moderately resistant (MR, 15 < DI ≤ 30), susceptible (S, 30 < DI ≤ 45), and highly susceptible (HS, DI > 45). Six varieties/lines were found resistant (R) to CR and 79 varieties/lines were moderately resistant (MR) (Table 2 and Appendix A). Resistant varieties/lines, including cultivars Zimai 12 and Xinong 509 and landrace Mazhamai from Huang-Huai winter wheat region, landrace Sifangmai from southwestern winter wheat area, landrace Huxumai from winter wheat region of the middle and lower reaches of Yangtze valley, and Canadian cultivar Dawson, accounted for 1.15% of the total tested varieties/lines. Among them, the DI of Zimai 12 and Huxumai was lower than the resistant control cultivar Sunco (DI = 11.5) in the two experiments. In addition, for the popularized cultivars Xibei612, Liaomai 16, Heng 6632, Zhengmai 9045, and Zhongyu 12 from Huang-Huai winter wheat region, cultivar Zhenmai 168 from winter wheat region of the middle and lower reaches of Yangtze valley, cultivar Mianyang 26, and landrace Yuqiumai from southwestern winter wheat region, cultivar Jichun 1016 from northeastern spring wheat region, cultivar SIRMIONE from Italy, cultivar OPATA and breeding line Ron2-Fnd×CMH74A.630 from Mexico, cultivars Owens, Freedom, and VAIOLET from the United States, and cultivar Aguilal from Moroccan, these varieties/lines resistance reached R or MR (close to R) in two years, and the DI was less than 25, accounting for about 3.07% (Table 2 and Appendix A).

### 2.3. SNPs Linked to CR Resistance Identified by GWAS

The results of genotyping showed that a total of 403,150 SNPs were detected after excluding low-quality loci, of which 396,807 loci were distributed on 21 chromosomes of wheat. The maximum number of SNPs distributed on B genome were 180,142, followed by A genome (160,050) and D genome (56,615). The length of the physical map covering A genome, B genome, and D genome was 4934.50 Mb, 5179.83 Mb, and 3950.66 Mb, respectively, and SNP densities were 32.43 SNP/Mb, 34.78 SNP/Mb, and 14.33 SNP/Mb, respectively. Among all the chromosomes, chromosome 3B had the highest marker density (53.72 SNP/Mb) and chromosome 4D has the lowest marker density (7.87 SNP/Mb). The average value of polymorphism information content (PIC) of all chromosomes ranged from 0.25 to 0.32, in which the PIC of 5B was the highest and that of 4A was the lowest. The PCA diagram (Figure 2a) visually shows the structural stratification of the samples. Some samples that significantly deviate can be removed and reanalyzed. In the subsequent GWAS analysis, the position information on these PC axes can be corrected as covariates in the regression analysis. Based on Bayesian algorithm in STRUCTURE2.2.3, the range of K value was set to 2–8. When K = 8522 wheat cultivars/lines were divided into 8 subgroups, which was the optimal population structure division (Figure 2 and Appendix A). 

LD (genome linkage disequilibrium) analysis showed that the LD attenuation distance of B genome was the largest, at 5.4 Mb. The attenuation distance of D genome was the lowest, at 1.6 Mb. The attenuation distance of A genome between B and D was 2.8 Mb. In addition, the attenuation distance of the whole genome was 3.4 Mb (Figure 3).

GWAS results (Figure 4a and Table 3) showed that 32 SNPs were significantly correlated with seedling resistance to CR in 2019 (*p*-value < 10^−4^), located on chromosomes 1A, 1B, 1D, 2A, 3A, 4D, 6D, and 7A, respectively; 127 SNPs were significantly correlated with seedling resistance to CR in 2020, located on chromosomes 1A, 1B, 1D, 2D, 4A, 4B, 4D, 5A, 5D, 6B, and 7B, respectively (Figure 4b and Table 3). Integrating the GWAS results (Table 3) of two years showed that two SNPs, namely AX-94897165 and AX-95121113, were detected on chromosome 1A. Twenty-three SNPs, namely AX-109317481, AX-111771935, AX-110379190, AX-94871733, AX-111634558, AX-94548073, AX-111818830, AX-94843699, AX-94428697, and so on, were detected on chromosome 1B during two years. In addition, significant SNPs, namely, AX-94799254 on chromosome 1D, AX-109468542 on chromosome 4A, and AX-95120025 on chromosome 4D, were also detected in two years. Among them, one SNP, AX-94897165, was located on chromosome 1A, and nine SNPs, AX-109317481, AX-111771935, AX-110379190, AX-94871733, AX-111634558, AX-94548073, AX-111818830, AX-94843699, and AX-94428697, were located on chromosome 1B, and had a strong correlation with disease resistance phenotype (*p*-value < 10^−5^ in two years); phenotypic variances explained (PVEs) by these SNPs were more than 8.45%. These results show that there are relatively stable CR resistance genes at seedling stage in the vicinity of these SNPs detected.

## 3. Discussion

CR is becoming a serious threat to wheat production in recent years, and there is a lack of available resistance sources and resistance genes to tackle CR. *F. pseudograminearum* can survive in stubble for up to three years [10], but until recent years, CR was considered to be the main disease threatening wheat production in China, so historically only limited efforts have been made to screen wheat for CR resistance [20]. The planting methods of retaining soil moisture through no-tillage strategy and stubble in the fields are the main reasons for the increase in the incidence of CR in many parts of the world [11]. For the prevention and control of CR, there are mainly some conventional measures, such as reducing straw returning to the field, and limiting the amount of fungi to spread and so on [32,33,34,35,36]. Although these dealings can reduce the disease, they cannot solve the problem completely. Rational utilization of disease-resistant germplasm resources is the main way to reduce the occurrence of diseases effectively and in an environment friendly manner.

At present, the wheat germplasm immune to CR has not been found, and only a few germplasms which are controlled by slightly effective polygene have moderate resistance to CR [5]. Yang et al. identified 88 wheat varieties from the Huang-Huai region of China in 2015 and found that all tested varieties had no disease resistance at seedling stage [37]. In addition, Yang et al. evaluated 234 wheat varieties in the same area for many years and found that nearly 83% of the varieties were sensitive to CR, and only 7 varieties were resistant to disease [7]. Increasing the scope of resistance source selection, exploring excellent resistance sources and resistance genes, and cultivating varieties with stable resistance are the urgent requirement to overcome the difficulties of wheat CR. In this paper, based on the screening results of varieties/lines resistant to CR in two years, we found that seven varieties/lines were resistant (R) to CR and 80 varieties/lines with moderate resistance (MR) to CR were stable (Appendix A). Among all the tested varieties/lines, landrace accounted for only 11.49%, but the proportion of landrace in varieties/lines resistant to CR reached 42.86%. Jin et al. also found that most of the varieties/lines including China’s main varieties are highly sensitive [20]. This shows that there are excellent resistance sources in landrace. This conclusion is similar to the conclusion drawn earlier in the identification of barley CR resistance [38]. Therefore, in the long-term breeding of varieties, breeders did not screen CR resistance for wheat or ignored the effect of CR on varieties, resulting in fewer resistance sources in cultivars, while more resistance sources were retained in landrace. Furthermore, in the screening results of CR resistance, we found that the proportion of semi-winter wheat varieties/lines in resistant varieties/lines was much higher than that of spring varieties/lines and winter varieties/lines. The possible reason is that semi-winter varieties/lines reduce the initial infection of pathogens by avoiding diseases in time and space. Wheat CR-resistant genetic research and breeding effort is now focused on collecting wheat germplasm reflecting ecological types for resistance source identification and exploring resistance genes.

In this study, GWAS technique was used to identify the distribution of SNPs related to CR on chromosomes 1A, 1B, 1D, 4A, and 4D, which further indicated that wheat CR resistance was a quantitative character controlled by minor polygenes, and the resistance loci involved multiple chromosomes. In addition, the significant loci AX-110436287 on chromosome 1B and AX-94692276 on 1D are closer to the lociAffx-88612017 and Affx-109205872, respectively, found by Jin et al. (2020) (physical distance < 1 Mb). The significant loci AX109621209 are closer to the *Xgwm11* on chromosome 1B (physical distance < 1.52 Mb) found by Martin et al. [23]. At present, our research group is verifying and fine-mapping these SNP loci through the linkage population in order to provide effective information for cloning disease resistance genes and breeding disease resistant varieties. 

## 4. Materials and Methods

### 4.1. Pathogen and Wheat Materials

Highly pathogenic *F. pseudograminearum* strain SX4-6 samples were isolated from the diseased fields in the Fuping County, Shaanxi Province, by the Soil-borne Disease Research Group of the College of Plant Protection, Northwest A&F University, Shaanxi-China. A total of 522 wheat varieties/lines representing genetic diversity were provided by the State key Laboratory of crop stress Biology for Arid area of Northwest A&F University, Shaanxi, China, divided into 437 cultivars, 60 landrace and 14 high-generation breeding lines, and 11 unknown varieties. Sunco, a control variety resistant to CR, was provided by the National Wheat Engineering Technology Research Center of Henan Agricultural University, Henan-China.

### 4.2. Seedling Resistance Assessment

A proper amount of washed wheat seeds was boiled for 30 min, then they were placed on Kraft paper, to dry the surface water, and packed into 500 mL culture bottles, sterilized at 120 °C for 30 min, then taken out and cooled after drying for 3–5 h. A small amount of isolated and purified strain SX4-6 was cultured on PDA (potato dextrose agar) medium and placed under shade at 25 °C for 3–4 days, so that the PDA medium was covered with fungal hyphae. The activated fungal medium blocks were mixed into wheat grain medium in a super clean worktable and cultured at 22 ± 1 °C for 14–21 days. In the process of culturing, the culture flask was shaken irregularly every day to make the mycelium grow well, and the amount of mycelium carried by the wheat grain medium was made consistent by transferring the inoculated wheat grains. Wheat seeds, flowerpots (about 14.5 cm in diameter), and trays were disinfected before use. Fifteen wheat seeds were planted in each pot, and two pots (two replications) were planted in the field. At the second leaf stage of wheat, 3–4 inoculated wheat grains were placed close to the base of each wheat stem and covered with a layer of soil and water, and moisturized for 3 dpi (days post inoculation). After that, the amount of watering reduced [30]. The disease was investigated at 30 dpi, and the disease was classified by 0–6 grade standard [19]. Disease index (DI) was calculated, and we utilized IBM SPSS Statistics software version 24 (https://www.ibm.com/cn-zh/products/spss-statistics/details, accessed on 29 May 2022) to analyze the variance of DI during two years (2019 and 2020) for detecting whether there was a significant difference in incidence. All varieties/lines were screened and grouped according to the overall distribution of DI. To exclude the influence of the environment on the phenotypic results of the seedling stage, we recorded the weather conditions in the test month.

### 4.3. Genotyping Analysis

A total of 522 wheat varieties/lines were analyzed by 660KSNP microarray. Affymetrix Genotyping ConsoleTM (GTC) software version 4.1 was used to assess the genotypic data. Low-quality SNPs with >10% missing values and with major allele frequencies of >95% were removed. Then, we conducted subsequent genome-wide association analysis of the remaining SNPs. In addition, polymorphism information content (PIC) was utilized to reflect the diversity of ectopic loci [39].

### 4.4. Population Structure, Linkage Disequilibrium, and Genome-Wide Association Analysis

The software STRUCTURE 2.2.3 was utilized to calculate the population structure [40]. The burn-in operating parameter was set to 10,000 and Monte Carlo Markov chain (MCMC) was set to 20,000. The range of K value (subgroups number) was determined to be 2–10 according to ΔK, repeated 5 times. The R language was utilized to make the genetic relationship heat map. The genotypic data of 522 wheat varieties/lines were analyzed by whole-genome and sub-genome linkage disequilibrium analysis (LD) by PLINK software version 2 (http://www.cog-genomics.org/plink2, accessed on 6 May 2022) [41]. The reference standard of LD is the correlation coefficient R2 between SNPs after screening. The parameters of calculating r2 were set to r2–ld-window-kb 30,000–ld-window 1000–ld-window-r20. The LOESS curve was drawn according to the calculated results, and the distance between markers with R2 ≤ 0.1 was taken as the standard of genomic LD attenuation distance. The resistance phenotypes and genotypes of 522 wheat germplasms were analyzed by GEMMA software, and the analysis model was based on univariate mixed linear model (mlm), general linear model (glm), and fixed and random model circulating probability unification (FarmCPU). The range of recommended *p*-values of each chromosome was calculated by GEC (Genetic Type I Error Calculator) software (http://pmglab.top/homepage/, accessed on 14 May 2022), and the threshold of significant loci was 10-4. The Manhattan map and QQ map were drawn by QQman software package (https://www.r-project.org/ accessed on 2 June 2022), and the heritability was calculated by GCTA software (https://yanglab.westlake.edu.cn/software/gcta/#Download, accessed on 15 June 2022).

## 5. Conclusions

In conclusion, we identified and evaluated the CR resistance of 522 wheat varieties/lines representing genetic diversity, screened the excellent resistance sources, and identified the CR resistance loci of wheat by genome-wide association analysis. Six varieties, including Zimai 12, Xinong 509, Mazhamai, Sifangmai, and Dawson, were classified as resistant (R) to CR. Seventy-nine varieties were classified as moderately resistant (MR) to CR (Appendix A). One SNP, AX-94897165, was located on chromosome 1A, and nine SNPs, AX-109317481, AX-111771935, AX-110379190, AX-94871733, AX-111634558, AX-94548073, AX-111818830, AX-94843699, and AX-94428697, were located on chromosome 1B, and had a strong correlation with disease resistance phenotype. These results enriched the diversity of CR resistance resources and laid a foundation for genetic improvement for CR resistance in wheat variety development programs in the future.

## Figures and Tables

**Figure 1 plants-11-01912-f001:**
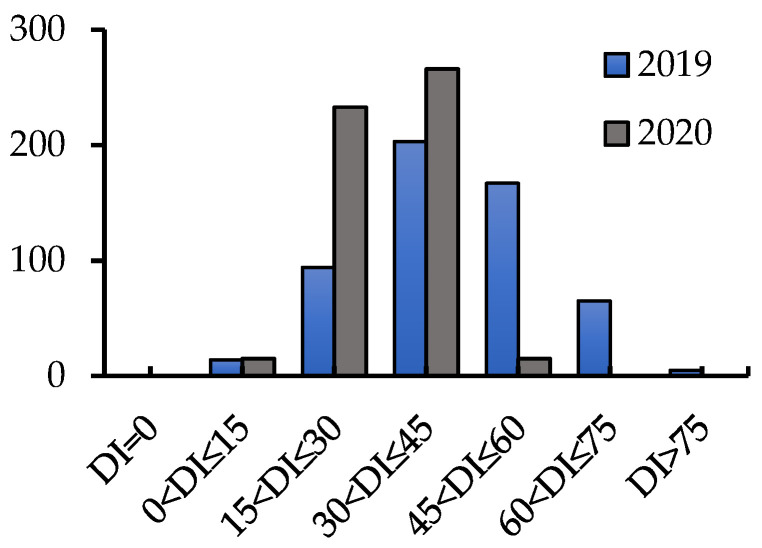
Phenotypic distribution for CR resistance in two years.

**Figure 2 plants-11-01912-f002:**
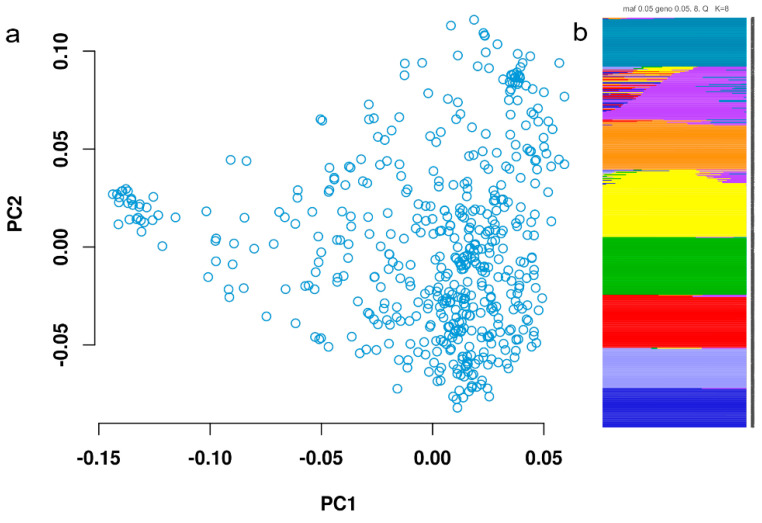
Principal component analysis 3D diagram and population structure diagram. (**a**) Principal component analysis. Each point in the PCA diagram represents a sample, and the information used in the drawing is the position of these samples on the two axes of PC1 and PC2. (**b**) The optimal population structure. Different colors represent different subgroups.

**Figure 3 plants-11-01912-f003:**
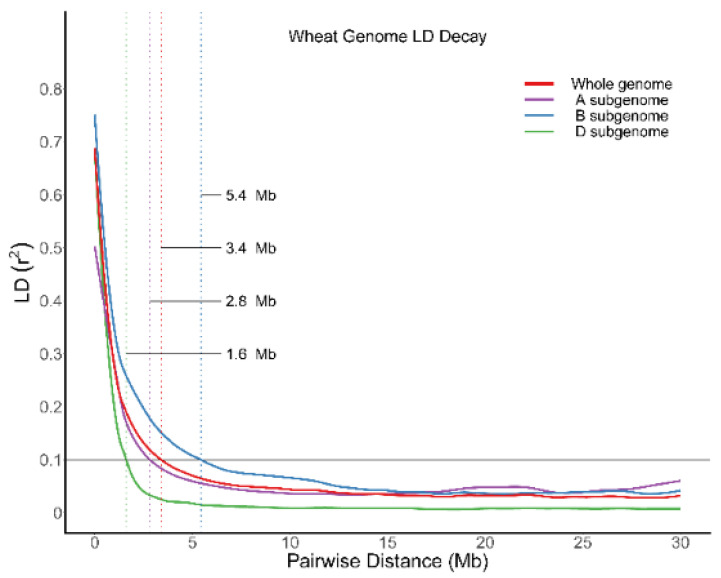
Genome linkage disequilibrium (LD) attenuation map (LD coefficient (r^2^) greater than 0.8 indicates that SNP is strongly correlated with functional mutation. If LD coefficient is less than 0.1, it can be considered that there is no correlation).

**Figure 4 plants-11-01912-f004:**
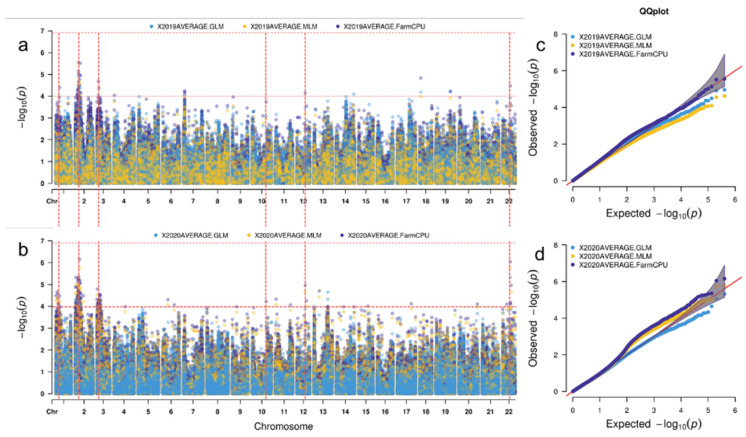
Genome-wide association analysis of CR resistance. Manhattan map and QQ plot map (**a**); (**c**) sowing time is 2019; (**b**,**d**) sowing time is 2020. The *X*-axis and *Y*-axis of QQ graph represent −log (*p*-value) of each SNP. The predicted line is a dotted line with an angle of 45° from the origin. Its deviation indicates that the actual value deviates from the predicted value. The SNP point has a large deviation, so it is considered that the deviation of the observed value of this SNP site is caused by the genetic effect caused by this SNP mutation.

**Table 1 plants-11-01912-t001:** ANOVA for wheat reactions to CR in 2019–2021.

Years		SS	DF	MS	F Value	*p*-Value
2019	Cultivar	195,492.43	521	375.23	11.109	<0.0001
	Replication	38.72	1	38.72	0.18	0.67
	Error	15,672.93	464	33.78		
	Total	211,204.08	986			
2020	Cultivar	93,754.55	521	179.95	14.5	<0.0001
	Replication	180.53	1	180.53	1.88	0.17
	Error	6478.14	522	12.41		
	Total	100,413.22	1044			

SS square sum; DF degrees of freedom; MS mean square.

**Table 2 plants-11-01912-t002:** Varieties/lines with fine resistance at both years.

Years	2019		2020	
Varieties	DI	Resistance	DI	Resistance
Zimai12	6.7	R	15	R
Huximai	10.8	R	10.4	R
SIRMIONE	11.7	R	16.7	MR
Xinong509	11.8	R	13.9	R
Sifangmai	13	R	12.5	R
Dawson	13	R	14.2	R
Mazhamai	13.5	R	11.1	R
ROMANIAN	14.8	R	14.6	R
Zhengmai9405	14.9	R	19.4	MR
Yuqiumai	15.2	MR	11.5	R
OPATA	15.2	MR	20	MR
Xibei612	15.2	MR	17.4	MR
Liaomai16	15.5	MR	18.3	MR
Owens	16.7	MR	18.1	MR
Aguilal	18.2	MR	16.4	MR
Heng6632	18.5	MR	18.3	MR
Ron 2-Fnd×CMH74A.630	18.6	MR	15.8	MR
Mianyang26	19.2	MR	16.7	MR
Zhenmai168	19.4	MR	12.9	R
VAIOLET	19.8	MR	18.1	MR
Zhongyu12	20.1	MR	17.4	MR
Ailiduo	20.8	MR	16.8	MR
Jichun1016	22.5	MR	18.1	MR

**Table 3 plants-11-01912-t003:** SNPs significantly associated with both years.

Snps	Chromosomes	Position	Allele		P-2019	P-2020
AX-94897165	1A	149510383	G	C	3.99 × 10^−^^5^	3.36 × 10^−^^5^
AX-95121113	1A	130909578	C	G	1.09 × 10^−^^4^	9.73 × 10^−^^5^
AX-109317481	1B	130743321	A	G	2.81 × 10^−^^6^	1.97 × 10^−^^5^
AX-111771935	1B	192551773	A	G	3.07 × 10^−^^6^	8.73 × 10^−^^6^
AX-110134543	1B	131236241	T	C	7.28 × 10^−^^6^	1.11 × 10^−^^4^
AX-109373335	1B	134968473	A	C	9.47 × 10^−^^6^	1.06 × 10^−^^4^
AX-110379190	1B	196926881	A	C	1.11 × 10^−^^5^	4.13 × 10^−^^5^
AX-94871733	1B	172946932	T	C	1.67 × 10^−^^5^	7.04 × 10^−^^7^
AX-111634558	1B	176745047	T	C	3.23 × 10^−^^5^	4.87 × 10^−^^5^
AX-94548073	1B	135938256	A	C	3.48 × 10^−^^5^	4.40 × 10^−^^6^
AX-111818830	1B	93459336	T	G	5.06 × 10^−^^5^	7.24 × 10^−^^6^
AX-94843699	1B	120884018	A	G	5.58 × 10^−^^5^	5.65 × 10^−^^6^
AX-94428697	1B	113516152	C	T	8.54 × 10^−^^5^	2.00 × 10^−^^5^
AX-111040346	1B	162264806	G	A	1.08 × 10^−^^4^	9.07 × 10^−^^6^
AX-111533020	1B	104346417	C	A	1.11 × 10^−^^4^	5.38 × 10^−^^5^
AX-110436287	1B	6798183	T	C	1.15 × 10^−^^4^	1.30 × 10^−^^4^
AX-94860874	1B	123352247	G	C	1.19 × 10^−^^4^	5.38 × 10^−^^6^
AX-108876206	1B	207814115	G	A	1.25 × 10^−^^4^	1.40 × 10^−^^4^
AX-111186969	1B	187708395	T	C	1.26 × 10^−^^4^	6.66 × 10^−^^6^
AX-95201485	1B	120884153	G	T	1.31 × 10^−^^4^	1.84 × 10^−^^5^
AX-94850477	1B	120883952	T	G	1.34 × 10^−^^4^	9.92 × 10^−^^5^
AX-111081045	1B	93564840	A	C	1.39 × 10^−^^4^	6.15 × 10^−^^5^
AX-95083508	1B	120884060	T	C	1.45 × 10^−^^4^	5.65 × 10^−^^6^
AX-109621209	1B	217554339	A	C	1.72 × 10^−^^4^	2.70 × 10^−^^5^
AX-109981169	1B	113652092	T	C	1.89 × 10^−^^4^	6.49 × 10^−^^5^
AX-94799254	1D	90633472	T	C	1.14 × 10^−^^4^	2.74 × 10^−^^5^
AX-95232541	1D	40579315	A	G	1.14 × 10^−^^4^	9.13 × 10^−^^5^
AX-94445466	1D	123109328	A	G	1.44 × 10^−^^4^	2.99 × 10^−^^5^
AX-94818806	1D	73729963	G	A	1.48 × 10^−^^4^	5.20 × 10^−^^5^
AX-94692276	1D	130308958	A	C	1.52 × 10^−^^4^	9.80 × 10^−^^5^
AX-110415885	1D	580926	C	T	1.79 × 10^−^^4^	4.88 × 10^−^^5^
AX-109468542	4A	539162489	T	C	1.82 × 10^−^^4^	6.06 × 10^−^^5^
AX-95120025	4D	329008071	A	G	7.05 × 10^−^^5^	1.10 × 10^−^^5^

## Data Availability

Not applicable.

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
