# Peer review of "Source Identification and Genome-Wide Association Analysis of Crown Rot Resistance in Wheat"

_plants, 2022, doi:10.3390/plants11151912_

Round 1

Reviewer 1 Report

In this study, 522 wheat varieties were evaluated for CR resistance and GWAS analysis was performed using the wheat 660 K genotyping assay. This study provided information for utilization of CR-resistance wheat germplasm and further explored resistance related genes, may lay a foundation for genetic improvement and variety breeding for wheat resistance to CR.

In line 177, do you think if the threshold (10-4) is too loose? Bonferroni Correction has been widely used to determine the threshold. Do you think if the threshold determined by Bonferroni Correction is more reasonable than the threshold (10-4) in this study?

Only one GWAS method was used to analyze the data. Do you think if it is more reliable to add more widely used GWAS methods as comparisons in data analysis? The loci detected by multiple GWAS methods may be more reliable to associate with the target trait.

The GWAS analysis were performed separately for 2019 and 2020. There are loci detected only in one year. Do you think if the potential environment and gene interactions may be an important role in this study? Could you add more details about the environment for the two years?

Reviewer 2 Report

The manuscript by Pu et al describes a wheat germplasm screen coupled with a GWAS for resistance to Fusarium Crown Rot. The phenotypic screen conducted across two years appears quite robust which is absolutely critical to this type of work being able to identify accessions showing resistance in both years and somewhat similar distributions for disease indexes across the two years. While the GWAS analysis is certainly a valid approach in identifying components of resistance to disease in wheat, it seems to me that for a disease like crown rot where the genetics appears to be very complicated the outputs of the phenotypic screen might be best utilised or validated in bi-parental mapping populations rather that GWAS. While I have never been involved in GWAS work, it did not appear to me that the identified markers associated with FCR resistance were particularly well supported statistically calling into question whether a GWAS approach is the correct one to take for this particular disease. If this is the case then perhaps this is a more important finding of the work rather than finding of unique but poorly supported loci. Overall the work, to the best of my ability, appears technically valid but I found the manuscript to be a little below par in terms of polish on the figures and written English. I have a few more specific queries below.

Lines 49-51. Not all of these varieties are considered resistant to Fusarium crown rot.

Line 82-84. This sentence around the adoption of GWAS to plant populations is very strangely structured and difficult to read

Figure 1: this should be a histogram not a line plot

Lines 142-150: This section on the cultuvars origin is difficult to read and perhaps would be better added to the table

Sowing times: Throughout the manuscript the terminology sowing times is used. This is probably better reserved for experiments where perhaps early and late plantings are conducted. I think the authors mean “year” not “sowing time”

Line 157-159: how does the physical map length come from a GWAS? This probably needs referencing where this information came from

Figure 2: The legend to this figure is extremely short and it is not clear at all to me what the PCA is showing with respect to the population understudy.

Figure 3: Again the legend is extremely short and it is not clear what this means in the context of the population being analysed. Is this typcialy for a GWAS with wheat? The figure legends in a paper should be somewhat stand alone from the rest of the paper but neither the text in the paper or the figure legend really explain what this means.

Lines 181-188: Listing all the SNPs here makes it very difficult to read and perhaps could be better just left to the table

Figure 5: again the figure legend is lacking details. The QQ plot needs explaining. To my reading the Manhattan and QQ plots like these would suggest the study is not particularly powerful in identifying associations. If this is the case then it needs to me made clearer

Line 228: It is unclear currently if Fusarium crown rot pathogens co-evolved with wheat. Certainly for Fusarium graminearum this does not appear to be the case with this pathogen likely to have come from central or North America.

Lines 232-235: this seems a bit repetitive of the introduction

Line 266: bacterial? Cake

Line 269-270: this sentence doesn’t read very well

Section 4.3: This short section is particularly poorly written and difficult to follow what has been done

Section 5: it seems odd to put the conclusions after the methods in a paper where the methods come after the results and discussion

Reviewer 3 Report

Crown rot (CR) is one of the most important diseases of wheat. The current manuscript identified new resistance sources to CR and dissected the CR resistance in a panel of 522 wheat varieties/lines by GWAS. This study provided useful information for the utilization of CR-resistant wheat germplasm and further understanding of the molecular and genetic basis of CR resistance in common wheat. I advise acceptance subject to several minor revisions on the writing.

Here are some suggestions but not limited to these:

Line2 Suggested title: Source Identification and Genome-wide Association Analysis of Crown Rot Resistance in Wheat

Line19 change “two sowing times” to “two different years”?

Line20 How did the four SNPs on chromosome 1B show the most significant resistance to disease?

Line22 “closer” should be “close”

Round 2

Reviewer 2 Report

All my original comments have been addressed appropriately. It will be very interesting to see if the regions identified in this GWAS study can be verified in biparental populations but I acknowledge that is beyond the scope of the current work.